# Prandial Correlations and Structure of the Ingestive Behavior of Pigs in Precision Feeding Programs

**DOI:** 10.3390/ani11102998

**Published:** 2021-10-19

**Authors:** Bruna C. K. Gomes, Ines Andretta, Marcio Valk, Candido Pomar, Luciano Hauschild, Alícia Z. Fraga, Marcos Kipper, Luciano Trevizan, Aline Remus

**Affiliations:** 1Department of Animal Science, Universidade Federal do Rio Grande do Sul, Porto Alegre 91540-000, Rio Grande do Sul, Brazil; brunacrisgomes@gmail.com (B.C.K.G.); mar.kipper@gmail.com (M.K.); lucianoveterinario@hotmail.com (L.T.); 2Department of Statistics, Universidade Federal do Rio Grande do Sul, Porto Alegre 91540-000, Rio Grande do Sul, Brazil; marciovalk@gmail.com; 3Sherbrooke Research and Development Centre, Agriculture and Agri-Food Canada, Sherbrooke, QC J1M 0C8, Canada; candido.pomar@canada.ca (C.P.); aline.remus@canada.ca (A.R.); 4Faculty of Agricultural and Veterinary Sciences, Universidade Estadual Paulista “Júlio de Mesquita Filho”, Jaboticabal 14884-900, São Paulo, Brazil; luciano.hauschild@unesp.br (L.H.); aliciafraga@outlook.com.br (A.Z.F.); 5PEGASE, INRAE, Institut Agro, 35590 Saint Gilles, France

**Keywords:** behavior, feeding strategy, meals, precision nutrition, precision livestock farming

## Abstract

**Simple Summary:**

A better understanding of pig feeding behavior can provide critical information for improving feeding strategies, productivity, and animal well-being. However, the availability of information is very limited in this research field. Data collected using electronic feeders were used in this study to generate information on pig feeding behavior, such as the time, size, and duration of each visit and meal. Later, data were used to calculate prandial correlations that could be interpreted as hunger or satiety-regulating mechanisms. The results indicated that the hunger-regulating mechanisms were slightly stronger than the satiety-regulation mechanisms in the studied animals. A decrease in both regulating mechanisms was observed during animal growth. Feeding programs showed little influence on the feeding regulating mechanisms, with conventional feeding systems (a group of animals fed diets adjusted by phase) slightly differing from precision feeding programs (animals receiving diets adjusted daily to meet the individual nutritional requirements). The use of electronic feeders in research and field conditions is increasing, as well the variety of sensors available in the market. Thus, more studies focusing on pig feeding behavior must be developed in the coming years to further understand such a complex and interesting animal.

**Abstract:**

The feeding behavior of growing-finishing pigs was analyzed to study prandial correlations and the probability of starting a new feeding event. The data were collected in real-time based on 157,632 visits by a group of 70 growing-finishing pigs (from 30.4 to 115.5 kg body weight, BW) to automatic feeders. The data were collected over 84 days, during which period the pigs were kept in conventional (by phase and by group) or precision (with daily and individual adjustments) feeding programs. A criterion to delimit each meal was then defined based on the probability of an animal starting a new feeding event within the next minute since the last visit. Prandial correlations were established between meal size and interval before meal (pre-prandial) or interval after meal (post-prandial) using Pearson correlation analysis. Post-prandial correlations (which can be interpreted as hunger-regulating mechanisms) were slightly stronger than pre-prandial correlations (which can be interpreted as satiety regulation mechanisms). Both correlations decreased as the animals’ age increased but were little influenced by the feeding programs. The information generated in this study allows a better understanding of pigs’ feeding behavior regulation mechanisms and could be used in the future to improve precision feeding programs.

## 1. Introduction

Genetic selection, in addition to improving feed efficiency, may also affect feeding behavior as there is an association between those traits [1]. In this context, knowing the feeding behavior of modern pigs is key to determining its influence on productive performance. Detailed information on pigs’ feeding behavior is also useful when implementing precision feeding programs, in which patterns of feeder use by the pigs could be affected by the feed intake behavior of the species [2,3].

An important tool for understanding the structure of intake regulation is the analysis of prandial correlations, which can be defined as the relationship between the quantity of feed consumed in the meal and the pre- and post-meal-time intervals. In other words, prandial correlations are a means of understanding whether an animal regulates its intake based on how much it has just eaten or on how much time has passed since the last meal [4].

Significant pre-prandial correlations relate to the relationship between the quantity of feed consumed in the meal (meal size) and the length of the interval before the meal; this is interpreted as a satiety-related intake regulation mechanism [5]. On the other hand, post-prandial correlations relate to the relationship between the meal size and the length of the interval that follows it; this is interpreted as a hunger-related mechanism [5,6].

Few studies focus on prandial correlations, and they are often not standardized among themselves [2,4,7]. Further, many studies are not conclusive, mainly because they are developed from small databases (few animals and/or few observations per animal). With this in mind, large databases, constructed in real-time through electronic feeding systems, are of great value in understanding the structure of the animals’ feeding behavior.

The probabilistic approach to feeding behavior has previously been applied to birds [5] and dairy cows [8]. However, research on pigs is still scarce, and little is known about the structure of intake regulation and possible factors influencing feeding motivation in this species. We hypothesized that a previously proposed approach obtained with young growing birds and subsequently tested with large lactating ruminants could be applied for pigs to confirm whether the nutrition strategy interferes in terms of feeding behavior. In this context, the feeding behavior of growing-finishing pigs was analyzed to study prandial correlations and the probability of starting a new feeding event based on the circadian rhythm and age of the pigs, and the feeding program offered to them.

## 2. Materials and Methods

### 2.1. Data Source

The animals, facilities, and experimental procedures used in this study were previously described [3]. In brief, information on the feed intake of 70 pigs (35 female and 35 male), with the same high-performance genotype (Fertilis 25 × G-Performer 8.0, Genetiporc Inc., Saint-Bernard, Quebec, Canada), housed together in a single room was obtained using 5 automatic feeding stations (Automatic and Intelligent Precision Feeder, University of Lleida, Lleida, Spain). Each feeding station consisted of a single-access device in which helical conveyors simultaneously combined volumetric quantities of up to four feeds stored in separate compartments located on the top of the feeder [3]. Feed density was measured weekly, and the feeders were calibrated to provide the required amount of feed from feed volumes and ensure that the pigs individually received a feed blend with the desired nutrient concentration. Thus, the automatic feeders identified each pig that had a transponder inserted in its right ear and then blended and provided the feed after each demand from an animal.

Each demand by the animal (activated by a push-button) was processed by the feeder, which offered a service from 15 g at the start to 26 g at the end of the experiment. Each visit was, therefore, made up of various demands and, hence, the total feed consumed in the period in which the animal remained feeding.

The serving size needed to be a small portion to avoid situations in which the feed assigned to one pig is consumed by another animal if left behind in the feeder. However, it could not be so small that it would have discouraged the animal from performing its voluntary feed intake. For that reason, the size of the service was adjusted daily to be close to 120–140 services per pig per day following the manufacturer’s recommendation. Updating the serving size offered a means of keeping the number of demands as constant as possible during the growth of the animal. Using a fixed serving size during the entire project would have obligated the animals to make far more demands at the end of the project (i.e., when the feed intake was increased), which could have modified their feeding behavior.

The pigs generally left the feeder empty or with insignificant amounts of feed (less than the service size) after each visit, which ensured that each animal received the feed that was assigned to him. All the feeders were designed to provide meals to all animals, regardless of the feeding program. This feature allowed all animals to be housed in the same pen in a single group. Pigs had free access to all the feeders throughout the experiment. Only during rush hours could pigs have limited access to the feeders (the occupancy rate was usually around 90% from 16:00 to 18:00 h, but it was usually lower than 80% during the rest of the daylight period, while a 20% occupancy rate was observed during the nights). However, the average feeder occupancy rate was 55% and feeder occupancy seldom reached 100% [3].

Treatments were randomly assigned to pigs when they reached an average of 30.4 ± 2.2 kg body weight (BW). During the 84 days of the experiment, the pigs were housed in a single pen with a slatted floor in a mechanically ventilated room. The ambient temperature was gradually decreased from 22 to 18 °C when the animals reached about 100 kg live weight, thereby ensuring thermoneutral conditions. Fluorescent lighting was controlled by a timer switch and supplied from 06:00 to 18:00 h. Water was supplied with low-pressure nipple drinkers distributed all over the pen.

### 2.2. Diets and Feeding

Two experimental feeds (named A and B) were separately formulated based on net energy and standardized ileal digestible (SID) amino acids and using the same ingredient composition database (which analyzed the gross composition of ingredients derived according to the EvaPig software, version 1.3.1.4; INRA, Saint-Gilles, France), without growth promoters or any other additive. Feed A was formulated to meet the SID lysine requirements of the most demanding pig at the beginning of the growing period, and feed B was formulated to meet the SID lysine requirements of the least demanding pig at the end of the finishing period. The feeds were formulated according to standard recommendations for AA profiles [9] and digestible P [10]. Feeds A and B were produced in 1 batch each and were provided in the steam-pelleted form. The composition of the served feed was obtained by blending feeds A and B in order to meet the estimated SID lysine requirements of each pig each day using a real-time model [11]. The requirements for amino acids other than lysine were assumed to be proportional to the lysine requirements, based on the ideal protein profile proposed [9].

Five feeding programs (treatments) were tested in this study. The control treatment (conventional, CON) consisted of a program made up of three 28-day feeding phases. In each phase, the diet supplied to the animals in this group was obtained by blending fixed proportions of feeds A and B. The blend of each feeding phase was determined during the first three days of the phase to meet the requirements of the 80th-percentile pig in the population, as previously recommended for the maximization of population body weight gain [12]. The other feeding programs provided a blend of feeds A and B, the proportions of which were adjusted to provide 110% (MP110), 100% (MP100), 90% (MP90), or 80% (MP80) of the estimated SID lysine requirements of each pig each day.

### 2.3. Data Collection and Management

The feeders were equipped with a monitoring device that continuously and automatically recorded the start and end time of each visit of each pig (day, hour, minute, and second), the number of services, and the total amount of feed served in each visit. All the information collected was used to construct an initial database, which was managed and analyzed using R Statistical Software (version 2.14.0; R Foundation for Statistical Computing, Vienna, Austria).

In the first information filtering procedures, certain atypical observations were detected through graphical analysis and critically evaluated from a biological point of view. The data on animals with between-visit intervals longer than 720 min (12 h; 1.4% of the original database observations) were removed. Such episodes were probably related to atypical physiological conditions such as subclinical infectious or inflammatory processes or accidental loss of the identifying transponder. In the latter case, as soon as the loss was noticed, the animal received a new transponder and returned to the experiment. In both situations, the data were excluded from the database because they were considered without interest for this study (atypical conditions). Additionally, visits longer than 40 min and those shorter than 0.05 min were discarded (5 observations). These data were likely generated by a technical failure of the system. However, this was considered a one-off limitation, given that a fairly limited number of observations were removed from the dataset.

### 2.4. Definition of the Meal Criterion

Each animal visit to the feeder was recorded from the time the pig was detected by the feeder until the feeder stopped detecting the presence of the animal. However, the definition of a visit differs from the concept of a meal. In this study, a meal was considered to be a group of several visits that were interrupted by short breaks for rest, water intake, a pause, or a feeder change [2,13].

To group the visits in a non-arbitrary manner, a preliminary estimate of the probability of the animals starting a new feeding event within the next minute (*p*_start_) was calculated in relation to the time since the last visit to the feeder. Defining T as the random variable "time taken for an animal start a new feeding event since the last visit to the feeder (minutes)" and t as "a given time (minutes)", *p*_start_ can be expressed as
*p*_start_ = *p*(t ≤ T ≤ t + 1/T > t) = *p*(t ≤ T ≤ t+1)/*p*(T > t).

This procedure was performed based on a method that was previously described (Method 3” available in: 14) and recommended for situations where neither the within- nor the between-meal interval distributions are known. The *p*_start_ function is plotted against the interval between visits and the minimum point of the *p*_start_ curve is defined as the meal criterion [14]. A simple moving average (SMA) over five-minute intervals was applied to reduce the effect of these random variations on the *p*_start_ function.

### 2.5. Statistical Analysis

After the meal criterion was defined, a new database was created with visits grouped into meals. Descriptive statistics were used to characterize the variables before and after the use of the meal criterion. The data grouped into meals were also used to calculate the meal size (minutes) and the between-meal intervals (pre-and post-prandial, minutes).

A new analysis of *p*_start_ was also performed to establish the probability of starting to feed according to circadian rhythm, the effect of age, the effects of the feeding programs, and the prandial correlations following the procedures described above. The use of this method allowed the dynamics of the demand for feed over time to be observed, which made it possible to predict the probability of an animal returning to the feeder to start a new feeding event.

Finally, prandial correlations were obtained for the population and each animal using Pearson correlations between the variables ‘meal size’ and ‘interval before meal’ (pre-prandial) or ‘interval after meal’ (post-prandial). Analyses were performed using the COR function from the STATS package of R Statistical Software (version 2.14.0; Vienna, Austria). These correlations were analyzed according to the age of the animals, their circadian rhythms (day or night), and the different feeding programs used. Correlation coefficients were considered significant if *p* < 0.05.

## 3. Results

### 3.1. General Description and Meal Criterion

Throughout the experiment, the pigs consumed feed and gained weight according to the expected performance based on their genotype. No serious health problems were detected other than inflammatory hoof problems, which were unrelated to the treatments and identified in three castrated males during the last phase of the experiment. The animals involved were isolated from the group and received adequate veterinary treatment, and their data were not considered in the analysis. Detailed performance responses are available [3]. In short, feed intake and feed efficiency were not influenced by the evaluated feeding programs.

The first database contained information from 157,632 feeder visits (outlier data already excluded). In this dataset, the meal criterion value for the total experiment period was 37 min, and this was considered an appropriate meal criterion. A new database was obtained after the visits were regrouped into meals (successive first-demand visits occurring over no more than 37 min), which generated a set of 38,501 observations.

Application of the meal criterion significantly altered the descriptive statistics on feeding behavior responses as compared to the initial database (Table 1). This demonstrates the importance of correctly estimating the meal criterion as opposed to applying arbitrary criteria.

### 3.2. Probability of the Animal Starting a New Feeding Event

The probability of starting a meal increased according to the time interval since the previous visit (Figure 1), and this increase was different between light and dark hours. The difference between daytime and nighttime minutes was observed mainly as of 250 min when the probability of pigs starting to feed was higher during the night than during the day. It was also observed that up to approximately 400 min, both during the day and at night, the animals maintained a more homogeneous feeding behavior pattern, but this pattern changed significantly after that time.

The animals’ age was also a significant factor in analyzing the probability of starting a meal (Figure 2). For instance, younger animals (especially in the first 3 weeks of the experiment) showed higher probability values, and as the animals’ ages increased, those probabilities decreased.

The feeding programs differed little for the probabilities of starting a meal (Figure 3). However, it was possible to identify a grouping of the probabilities between the programs studied, to the effect that the conventional feeding program (by phase and by group) continued to show the lowest probability of starting a meal as compared to the precision feeding programs. The effect of sex was also tested, but did not result in significant variations; therefore, it was not presented.

### 3.3. Prandial Correlations

The pigs presented with slightly higher post-prandial correlations than pre-prandial correlations (Figure 4). The pre-and post-prandial correlation coefficients showed high variability between the individuals in the population, with some pigs having positive values while others showed negative correlations. This heterogeneity could indicate that the structure of the prandial correlations is a characteristic that is also influenced by factors that are intrinsic to the individual.

Both correlations (pre- and post-prandial) were higher in the first weeks and weakened gradually over time (Table 2). In addition, the pre-prandial correlations were significant (*p* < 0.05) only until the ninth week of the experiment. Pre-prandial correlations were greater during the night than during the day. In contrast, post-prandial correlations were higher during the day.

There was little difference between the feeding programs tested for prandial correlations (Table 3), which were slightly higher for the conventional program as compared to the precision feeding programs.

## 4. Discussion

### 4.1. Meal Criterion

Estimating feed intake by pigs is a complex task since feed ingestion depends on various intrinsic and extrinsic factors of the animal. Despite the difficulties, correctly estimating feed intake is fundamental for the implementation of more precise feeding programs. In addition, knowledge of the normal feeding behavior of pigs is fundamental when implementing automatic systems so that health disorders that are often associated with changes in feeding motivation can be identified.

To understand pigs’ feeding behavior, it is first necessary to carefully define meals within the database. The animals invariably take short breaks during a meal; because of this, several recorded visits to electronic feeders are commonly obtained. These events, although closely linked or separated by very short intervals of time (e.g., seconds or a few minutes), may be identified and recorded as more than one visit by the feeder, depending on its configuration. For proper assessment of the prandial structures, these short breaks (“intra-meal”) should be distinguished from the long breaks that take place between meals.

The methodology previously proposed [14] and used in this study is among the most accepted means of defining a specific meal criterion for the dataset. Despite the importance of this procedure, many recent studies have arbitrarily defined the meal criteria, i.e., without using an adequate methodology to identify the minimum time interval between meals. The impact of applying a meal criterion and even of choosing the method for defining this criterion on observed behavior responses cannot be denied, although it may be higher or lower depending on the database characteristics.

The meal criterion was estimated at 37 min in this study. Based on this criterion, visits by the same animal to the feeder, separated by intervals of less than 37 min (or 2040 s), were grouped as a single meal. The descriptive statistics differed strongly between the initial database (visits) and the revised database that took the meal criterion into account. This difference between the results obtained from the same database indicates that the use of a probabilistic model to define the meal criterion is of utmost importance. Using an inadequate criterion generally divides the meals into smaller portions [15], thereby interfering with the interpretation of the answers obtained from that dataset [16]. For example, when an arbitrary meal criterion was applied to rats [17], it was observed that food deprivation resulted in an increase in meal frequency without a change in meal size. However, when a meal criterion was estimated using an appropriate method, the results moved towards indicating that food deprivation increased in terms of meal size but not in terms of frequency.

Few studies on meal criteria in pigs are available in the literature. In a previous study [4], studies on pigs that proposed meal criteria varying between 1.2 and 30 min were listed. The high variation between-meal criteria used in the studies is probably due to the different methods used to estimate them. Using the same method employed in this study, meal criteria of 1200 s for broilers and 1725 s for Peking ducks were obtained [14]. Furthermore, a meal criterion that was previously found for Peking ducks was similar to that found in our study (2208 s, [18]). To our knowledge, this is the first time that a probability model has been used to define the meal criterion for pigs.

### 4.2. Description of Feeding Behavior

The feeding patterns of the pigs varied greatly within the population (between individuals). It is important to note that this heterogeneity in feeding behavior responses happened despite the animals being homogeneous for several characteristics, such as genetic lineage, origin, and age. In addition, all the animals were kept in the same stall and were exposed to similar conditions in terms of environment and handling. It is possible to infer, therefore, that individual characteristics (intrinsic factors of the animal) are a determining factor in the manifestation of feeding behavior.

In this study, the set of available data allowed for the analysis of changes in probabilities according to circadian rhythm, feeding program, age of the animals, and sex (non-significant). More studies are required to identify and understand the influence of other factors.

### 4.3. Probability of Starting a New Feeding Event—The Circadian Rhythm

The results of this study showed that during the night, there was a greater probability of the animals starting to feed after long intervals since the last meal. Previous methods [19] indicated that the probability of animals starting a meal is independent of the time elapsed since the last meal. However, according to the concept of satiety, when an animal is satiated at the end of a meal, the probability of starting a meal soon after is low but increases over time. In our study, this effect was most clearly identified in the nighttime period. However, during the day, it was observed only in the first minutes after the meal and was not clearly observed during the probability interval. The difficulty in establishing a pattern of behavior after long periods without a meal can be explained by the complex behavior of pigs, which remain more active during the day and divide their feeding time up amongst other activities, such as social and exploratory interactions.

### 4.4. Probability of Starting a New Feeding Event—Effect of Age

The probability of pigs starting to feed was higher in younger animals (i.e., in the early weeks of the experimental period). These results could be related to the more active profile of the young animals. In addition, the high rate of protein deposition associated with low capacity of the digestive tract, can be an important factor driving pigs’ feed intake behavior in the growth phase. Therefore, young pigs tend to go more often to the feeder to obtain the right amount of nutrients to meet their potential for muscle deposition [20].

### 4.5. Probability of Starting a New Feeding Event—Effect of the Feeding Programs

The probability of pigs starting to feed was less in the conventional feeding program (by phase and by group) as compared to other feeding programs (daily and individual). This could be related to the nutritional levels offered to the pigs in these feeding programs. The animals in the conventional program received diets with a higher concentration of lysine and other nutrients. Alterations in feed intake based on dietary supplementation with lysine were previously described in the literature [21]. In this study, although there was no difference in average feed intake in the feeding programs [3], it was possible to identify signs of changes in the feed intake regulation mechanisms that are likely related to the nutritional density of diets. These observations would be in line with classic intake regulation theories, such as the amino acid theory. Future studies are necessary to better explain this relationship and so that the specificities of feeding behavior can be taken into account when defining more precise feeding programs.

### 4.6. Prandial Correlations

When the objective is to establish a feeding program that considers the nutritional requirements of the pigs in a population context, knowledge of the variability between the animals is essential. This is because the average population response to a given feeding strategy may be different from the average individual response in the same population [11,12]. Thus, while it is no easy task, it is vitally important to consider the heterogeneity of populations in terms of performance and nutritional requirements. The high variation observed in the prandial correlations obtained in this study demonstrates that heterogeneity in terms of feeding behavior is also worth highlighting in future studies on precision nutrition.

Prandial correlations are important for better understanding the regulatory signals that are present in pigs’ metabolism. Generally speaking, meals cause a sensation of satiety in the animals. Therefore, the probability of an animal returning to the feeder after a meal should initially be low, but then increase as the satiety mechanism loses strength [5].

A high correlation between the meal size and the interval before the meal is expected when intake is controlled by satiety regulation mechanisms; i.e., the animal will start a meal when the satiety sensation falls below a critical point. When hunger control predominates, one expects to find a high correlation between the meal size and the interval after the meal.

In the current study, the observed correlations were relatively low, which suggests that other factors affect how animals regulate their feed intake pattern. Low correlations were also reported in previous studies carried out with other species [5]. This may also be related to the high number of individuals and observations in the database, as well as the natural variability of behavioral characteristics among the animals in the population.

The vast majority of pre-prandial correlations and post-prandial correlations identified in our study were significant. This differs from what was previously described [5], when only a small proportion of the variation in meal size was related to variation in the post-meal-time interval, with no effect of animal age on this response.

In this study, pre-prandial correlations were higher during the night, which suggests a greater influence of the satiety mechanism during this period. The pigs with higher pre-prandial correlations may have been unable to feed freely during the peak feeding activity time (daytime period) and, therefore, may have been forced to feed at times when there was less competition for the feeder (nighttime period). This hypothesis will need to be confirmed in future studies that consider the social dominance relationship not only as an influential factor in performance and feeding behavior, but also as a possible influential factor in prandial correlations and, consequently, in the way the animal expresses its intake regulation mechanisms.

Most studies with animals that feed ad libitum are regulated by satiety [5,22] and, in these cases, hunger mechanisms only play a role in specific situations. In our study, this predominance of satiety mechanisms was observed only during the nighttime period (which is equivalent to less time). In this context, it is important to consider that the type of prandial correlation may be a behavioral control measure of individuals that is influenced by factors such as species and social competition for access to the feeder [17,23].

Another aspect that should be highlighted is that the variation between individuals in the feeding pattern was lower in the first weeks of the experiment. This may indicate that variability between individuals also increases over time for feeding behavior characteristics. Therefore, even if all individuals are exposed to the same environment, they express different behaviors in a given situation.

## 5. Conclusions

This study falls within an area that is still very lacking in information. Therefore, although this is a predominantly theoretical study, the results obtained are of great value to researchers and nutritionists. The approach employed allowed us to identify which hunger mechanisms had stronger correlations during the daytime period. Moreover, the results indicated that the probability of starting to feed does not increase monotonically with time since the previous meal; instead, it presents characteristic behaviors in each post-prandial phase. If properly employed in future studies, the method of analysis proposed in this article could allow for a better understanding of the feeding behavior and prandial relations of pigs in different production and feeding conditions, and could contribute to the implementation of more precise practices when establishing feeding programs and methods for the early identification of disease.

## Figures and Tables

**Figure 1 animals-11-02998-f001:**
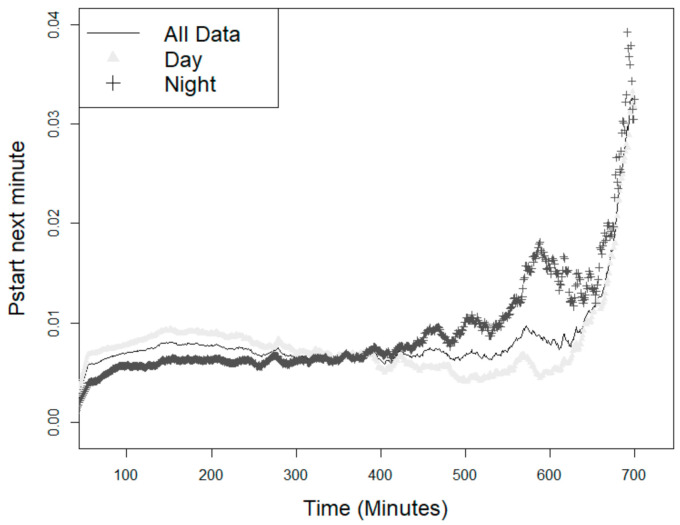
Probability of the animals starting a new feeding event (*p*_start_) within the next minute since the last visit—over the day, or during the light and night hours.

**Figure 2 animals-11-02998-f002:**
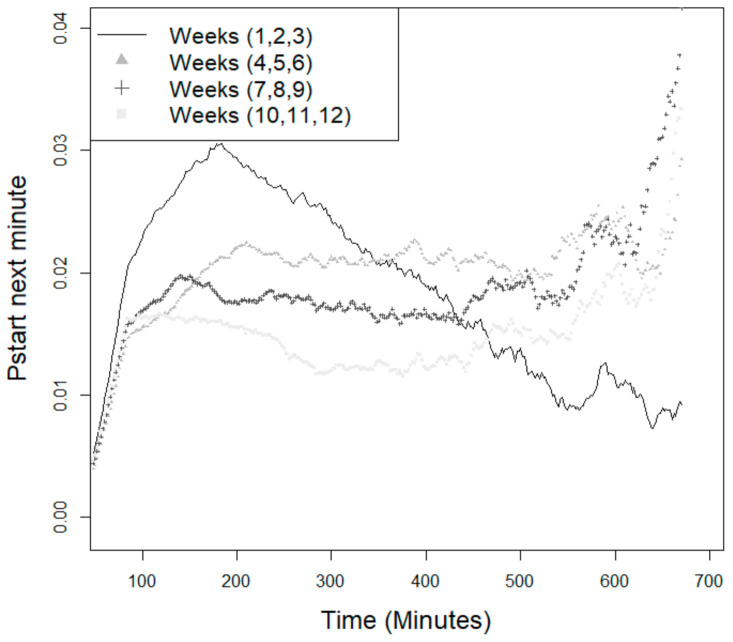
Probability of the animals starting a new feeding event (*p*_start_) within the next minute since the last visit—age effect.

**Figure 3 animals-11-02998-f003:**
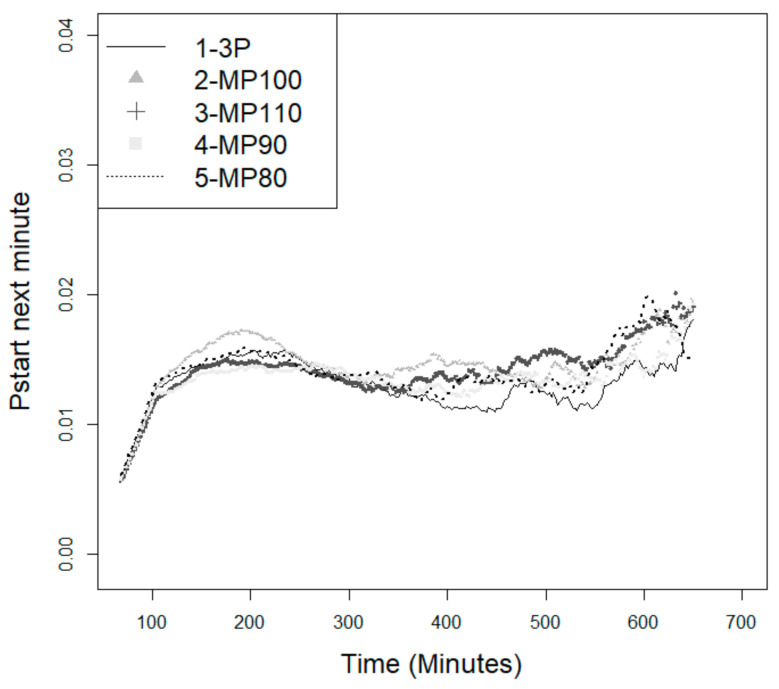
Probability of the animals starting a new feeding event (*p*_start_) within the next minute since the last visit—effect of feeding programs ^1^. ^1^ Three-phase feeding program provided by group (3P) or daily-phase feeding programs provided individually to meet 110% (MP110), 100% (MP100), 90% (MP90), or 80% (MP80) of the estimated nutritional requirements.

**Figure 4 animals-11-02998-f004:**
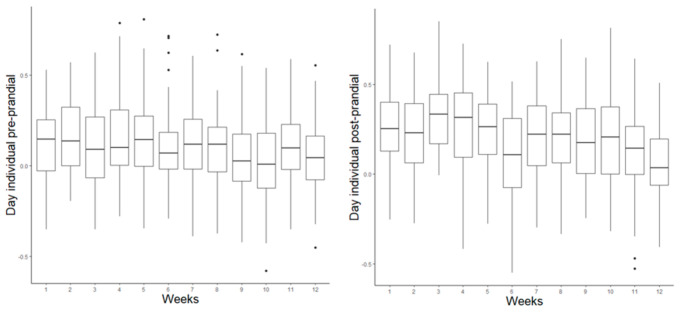
Pre- and post-prandial individual correlation coefficients were calculated as Pearson correlations between meal size and interval before (pre-prandial) or after (post-prandial) meals.

**Table 1 animals-11-02998-t001:** Feeding behavior of growing-finishing pigs raised from 30.4 to 100 kg body weight.

Response	First Quartile	Median	Mean	Third Quartile
Interval between visits, minutes	0.05	0.25	43.6	25.5
Time duration per visits, minutes	0.61	1.11	2.03	2.31
Intake per visits, g	21.0	44.0	89.5	104.8
Interval between meals, minutes	83.2	141.5	198.3	242.4
Meal size, minutes	5.05	9.33	12.95	15.65
Intake per meal, g	153.8	305.2	358.9	504.8

**Table 2 animals-11-02998-t002:** Weekly correlations ^1^ between the actual meal size and the interval between the actual and previous meal (pre-prandial) and between the actual and the following meal (post-prandial).

	Weeks											
	1	2	3	4	5	6	7	8	9	10	11	12
Pre-prandial correlations												
24 h	0.25 *	0.21 *	0.18 *	0.18 *	0.13 *	0.11 *	0.10 *	0.13 *	0.07 *	0.02 ^†^	0.02 *	–0.02 ^†^
Day	0.18 *	0.19 *	0.14 *	0.13 *	0.09 *	0.07 *	0.09 *	0.09 *	0.06 *	0.06 ^†^	0.02 *	0.03 *
Night	0.26 *	0.25 *	0.33 *	0.28 *	0.24 *	0.21 *	0.14 *	0.22 *	0.16 *	0.13 ^†^	0.14 *	0.15 *
Post-prandial correlation												
24 h	0.26 *	0.23 *	0.27 *	0.24 *	0.21 *	0.12 *	0.20 *	0.25 *	0.23 *	0.23 *	0.16 *	0.14 *
Day	0.32 *	0.28 *	0.35 *	0.30 *	0.24 *	0.13 *	0.22 *	0.27 *	0.22 *	0.24 *	0.17 *	0.14 *
Night	0.19 *	0.24 *	0.24 *	0.22 *	0.23 *	0.11 *	0.19 *	0.19 *	0.24 *	0.18 *	0.14 ^†^	0.13 ^†^

^1^ Estimates of pre- and post-prandial correlation coefficients were calculated as Pearson correlations between meal size and interval before (pre-prandial) or after (post-prandial) meal. * Significant and ^†^ non-significant correlation coefficient.

**Table 3 animals-11-02998-t003:** Pre- and post-prandial correlations according to feeding program.

	Feeding Program ^1,2^				
	3P	MP110	MP100	MP90	MP80
24 h pre-prandial	0.14	0.11	0.13	0.14	0.13
24 h post-prandial	0.26	0.23	0.23	0.20	0.21

^1^ Three-phase feeding program provided by group (3P) or daily-phase feeding programs provided individually to meet 110% (MP110), 100% (MP100), 90% (MP90), or 80% (MP80) of the estimated nutritional requirements. **^2^** All correlation coefficients were significant at *p* < 0.01.

## Data Availability

The data supporting the conclusions of this article will be made available by the authors upon request.

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
