# Peer review of "Prandial Correlations and Structure of the Ingestive Behavior of Pigs in Precision Feeding Programs"

_animals, 2021, doi:10.3390/ani11102998_

Round 1
Reviewer 1 Report
The experiment studied the relationship between meal intervals (defined using an artificially calculated meal criterion) with age, day-night, feeding program in growing/finisher pigs. Important topic. More details on computing the Pstart and meal criterion (37 mins) should be given.
My main concern is whether the age of pigs and the change of serve size (from 15-28 g per serve) affected the Pstart and therefore the calculated meal criterion. I guess the Pstart and meal criterion may be dynamic with the change of these two variables. For example, if you plot the Pstart against visits’ interval (Figure 1) by different ages, would you resolve a different meal criterion for each age group? If this is true, then the age effect has been confounded by the artificially determined meal criterion (fixed as 37mins). Wish authors could comment and clarify it in the method section.
Specific comments:
1. L72-74 Please state the hypothesis of the study.
2. L90-91 For the experimental design, could you please explain why the number of the feed services was controlled close to 120-140? Why not allow the pigs to demand more services at a fixed amount of feed per service?
3. L99 Please define when the “Rush hour” lasted?
4. L164 What is the unit of T and t? minutes? The mathematical equation for calculating Pstart should be given including the equations for calculating P(t<=T<=t+1) and P(T>t).
5. L166-169 Two methods were used for calculating meal criterion in Reference 14. Need to specify which one you chose to use and why not the other?
6. L166-169 How was the Pstart difference (see Figure 1 y-axis) calculated? Details on modeling the relationship of Pstart difference and visits intervals should be reported. How the timepoint that minimizes the Pstart difference was mathematically solved in the model should be reported.
7. L170-184 It is unclear what statistical method was used for analyzing the probability for starting the new feeding event and correlations? Statistical package information is missing.
8. Table 1 if the average interval between meals is 198 min, does it mean the probability of a pig to start a new feeding is 50% after 198 mins since last the meal? If this is true, why was the predicted probability only less than 0.01 at 200 mins (as shown in Figure 2)?
9. Why the probability of starting a new feeding event is so low? The maximum Pstart is below 0.05 in all figures. Does it mean it is unpredictable? Have I misunderstood the definition of Pstart?
10. Table 2 and Table 3 Which correlation is significant? Can you highlight the significant correlation with superscripts etc in the table?
11. All figures’ resolution is too low.
12. The x-axis in Figure 2, 3, and 4 ranged above 720 mins, but the method section stated (L141) that the visits with intervals longer than 720 min has been removed.
13. L218-220 as well as L335-341: was this finding confounded by the increased service feed with age? Compared with older pigs, younger pigs received less feed per serve and thus they had to increase the meal frequency.
14. L340-341 How does increasing feeding frequency change the ratio of nutrients if a single diet is offered? I guess the “ratio of nutrients” should be changed to the “amount of nutrients”.
Author Response
We appreciated all the comments provided. Your suggestions helped us to improve the quality of the manuscript and added value to the discussion.
Thank you very much for your time and attention!
All comments and questions were addressed and our answers are presented in the attached file. Additionally, we highlighted in red all changes made in the manuscript.

Reviewer 2 Report
Comments, review of manuscript id: Animals-1381784.
Title: ”Prandial correlations and structure of the ingestive behaviour of pigs in precision feeding programs”
The manuscript gives new information on the eating behaviour of growing-finishing pigs. The manuscript is well written, but need some minor corrections, as mentioned in the comments below.
The English language is good, but with some minor errors.
Comments.
Introduction, line 48. Here you write “…important treats, …”. Is “treat” the correct word to use. Or is this a misspelling? The word “traits” will probably be better.
Materials and Methods. 2.1. Data source, line 103. The housing of the animals is somewhat unclear. Were all 70 pigs housed in one single pen? If not, describe the number of pens used, and the number of pigs in each pen.
Materials and Methods. 2.4. Definition of the meal criterion, line 167. Spelling error, the correct is “plotted”, not “ploted”..
Figure 1. Check the lay-out of this figure. You have some text of the manuscript (line 199) between the figure and the figure legend.
Discussion. 4.5. Prandial correlations, line 381. Here you state “…with no effect of bird age…” this sounds confusing in a pig manuscript. If your statement refers to a study with birds, you should describe this better.
Author Response

(The authors gave the same response as above.)

Round 2
Reviewer 1 Report
Thank you for the reply to my comments.